# Remodeling of RNA-binding proteome and RNA-mediated regulation as a new layer of control of sporulation

Thomas Kaboré,[1] Maya Belghazi,[2] Christophe Verthuy,[2] Anne Galinier,[1] Clémentine Delan-Forino[1]

**ABSTRACT** Sporulation allows certain bacteria to survive extreme conditions for extended periods, posing challenges to public health and food safety. Regulation at the transcriptional level by σ factors has been well studied in the non-pathogenic model bacterium *Bacillus subtilis*, while post-transcriptional control remains poorly understood. RNA-binding proteins (RBPs) and small non-coding RNAs (sRNAs) modulate gene expression by affecting mRNA stability or translation. Recent studies suggest that RNA-mediated regulation plays a role in sporulation, but its networks and interactions remain largely uncharacterized, highlighting the need for further investigation. To address this knowledge gap, we adapted orthogonal organic phase separation (OOPS), a high-throughput method to specifically purify RNA-binding proteome (RBPome). By monitoring its dynamics, we observed that the RBPome is highly remodeled during *B. subtilis* sporulation process. Our approach led to the identification of novel RBPs and potential RNA-mediated post-transcriptional regulators of sporulation. This work provides important insights into the interplay between RNAs and RBPs, advancing the understanding of post-transcriptional regulation in Gram-positive spore-forming bacteria. It also offers new resources for exploring the molecular mechanisms that govern sporulation.

**IMPORTANCE** Understanding how bacteria survive extreme conditions is key to tackling challenges in health, food safety, and industry. This study reveals a previously unexplored layer of control in *Bacillus subtilis*, a model organism for spore-forming bacteria—those that can produce spores, which are highly resistant, dormant cells able to endure harsh environments. By mapping, for the first time, the full set of proteins that interact with RNA during sporulation, this work uncovers how bacteria fine-tune their internal programs. The identification of novel RNA-binding proteins sheds light on how bacteria adapt at the molecular level and lays a valuable foundation for future mechanistic studies that will deepen our understanding of bacterial adaptation and resilience.

**KEYWORDS** RNA binding proteins, UV crosslinking, sporulation, proteomics, *Bacillus subtilis*, adaptation, RBPome

B acteria can adapt to changing environments and survive extreme conditions. For example, under nutrient-limiting conditions, members of the *Bacillus* and *Clostridium* genera can undergo sporulation, differentiating into highly resilient dormant spores that enter extreme quiescence (1). When conditions improve, spores germinate and revert to metabolically active cells, resuming vegetative growth. They have been shown to do so even after 250 million years (2).

*Bacillus subtilis* is a rod-shaped bacterium ubiquitously found in soil and is one of the best-characterized Gram-positive spore-formers (3). *Bacilli* sporulation is a well-defined, multistage process (Fig. 1) (4). It begins with the formation of an asymmetric septum,

Address correspondence to Clémentine Delan-Forino, cdelan-forino@imm.cnrs.fr.

The authors declare no conflict of interest.

See the funding table on p. 15.

dividing the cell into a forespore (the future spore) and a larger mother cell that supports spore maturation.

Mechanisms controlling protein synthesis and enzymatic activity during sporulation have been widely investigated in *B. subtilis*, particularly the role of σ factors in orchestrating gene transcription (Fig. 1) (5–9). These alternative σ-factors direct the core RNA polymerase to different sets of promoters than those recognized by the housekeeping $\sigma^A$ (Fig. 1); they control more than 500 genes, most of which are not expressed during vegetative growth. Nearly a third of them remain functionally uncharacterized, and even those with assigned function in spore formation are mostly partially understood (10). To date, most research has focused on the characterization of "spo" genes whose deletion affects sporulation without disturbing vegetative growth (4). In contrast, post-transcriptional regulation of sporulation remains largely unexplored. It usually relies on dynamic interactions between RNA-binding proteins (RBPs) and their RNA targets, known to regulate the adaptive response across all domains of life by modulating RNA metabolism, cellular homeostasis, and stress adaptation (11–15).

In bacteria, RBPs often cooperate with regulatory RNAs such as small non-coding RNAs (sRNAs) to influence translation or transcript stability, mediating sRNA-mRNA interactions and/or recruiting degradation factors (16). Transcribed transiently in response to environmental shifts (17), sRNAs work with RBPs to target mRNAs to fine-tune various cellular processes (18, 19). Their involvement in sporulation is increasingly gaining attention (20, 21). For instance, the dual-function sRNA SR1, encoding a small peptide interacting with glyceraldehyde-3P-dehydrogenase, is also able to base-pair with kinA mRNA, encoding a kinase that controls sporulation in *B. subtilis* (20). Other factors such as CsfG, conserved among spore-formers, and KrrA, a conserved RBP identified in *Bacillus anthracis* (22, 23), regulate mRNAs involved in sporulation, germination, and competence.

Unlike Gram-negative bacteria, where general RBPs such as Hfq and ProQ facilitate sRNA-mRNA interactions (16), *B. subtilis* lacks ProQ, and Hfq appears to be dispensable for sRNA-mediated regulation (24–26). Instead, Gram-positive bacteria likely rely on distinct, process-specific RBPs, such as CsrA, a pleiotropic regulator that stabilizes SR1 interactions with *ahrC* but not *kinA* (27).

Emerging methods such as poly-A-independent RBP purification have facilitated RBP discovery in prokaryotes. For example, Grad-seq, which combines density gradient ultracentrifugation with high-throughput sequencing, led to the identification of ProQ (28). While powerful, this technique does not confirm direct RNA–protein interaction. UV irradiation has long served as a robust tool to capture direct protein–RNA interactions by inducing covalent crosslinking. UV-crosslinking-based methods, such as total RNA-associated protein purification (29), 2C (complex capture) (30), leverage the intrinsic affinity of RNA for silica, while others, like Phenol-Toluol extraction (31) and orthogonal organic phase separation (OOPS) (32), rely on organic extraction. These approaches have identified new RNA partners and regulatory pathways across diverse organisms and can isolate crosslinked protein–RNA complexes, enabling the study of RBP dynamics.

Here, we adapted OOPS for the first time to a Gram-positive bacterium, *B. subtilis*. We followed the dynamics of the RNA-binding proteome (RBPome) during sporulation and demonstrated that both RBP production levels and RNA-binding abilities are highly modulated throughout this process. Through this *in vivo* global approach, we spotted novel RBPs such as KhpA, a putative RNA chaperone, that may regulate sporulation.

## RESULTS

### Most putative independently transcribed sRNAs are specifically upregulated during sporulation

To assess the importance of RNA-mediated regulation during sporulation of *B. subtilis*, we first used available "tiling arrays" data (17), which identified 1,589 new RNA segments. Among them, about 150 are independently expressed, making them putative sRNAs. We represented their normalized expression profile during vegetative growth (in lysogeny

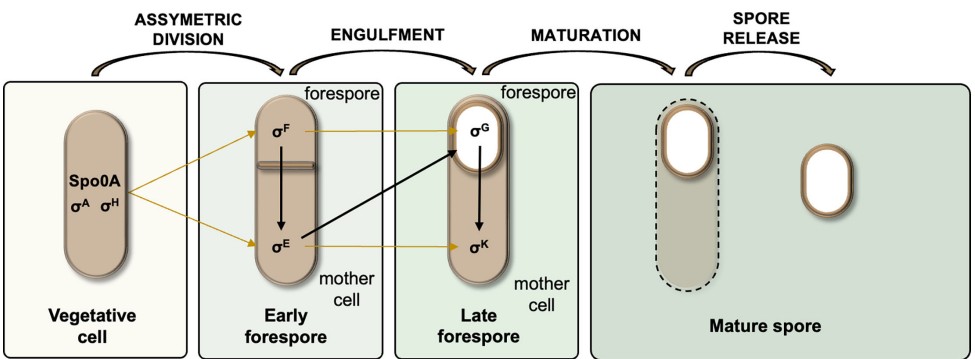

**FIG 1** Overview of sporulation with cell-specific (pre-spore or mother cell) σ transcription factors involved in each step (serial activation represented by arrows). σ factors are subunits of the RNA polymerase responsible for determining the specificity of promoter DNA binding and required for transcription initiation.

broth [LB] medium) and throughout sporulation as a heatmap (Fig. 2; Table S1). Over half of the independently transcribed RNAs are upregulated during sporulation, making it the tested condition with the highest number of overexpressed sRNAs (17). We then examined predicted σ factor dependencies (Fig. 2, right column, Table S1) (17, 33). Consistent with experimental data, 66 of 152 transcripts depend on sporulation-specific σ factors. These putative sRNAs show little or no expression during vegetative growth in LB. Notably, some non-coding RNAs not clearly associated with sporulation-specific σ factors are still enriched during sporulation, suggesting they play a role in the process, even if not exclusively involved in spore formation.

We observe that most sporulation-specific putative sRNAs are expressed under control of $\sigma^E$ and $\sigma^F$ (early sporulation stages), or under control of $\sigma^K$ (late mother cell) (Fig. 1), between 4 and 7 h of growth in sporulation medium (Fig. 2). These patterns suggest a regulatory role for sRNAs during sporulation.

## RNA-protein complexes stabilized by UV crosslink can be purified by orthogonal organic phase separation in *B. subtilis*

To identify proteins involved in RNA-mediated regulation during sporulation, we aimed to globally capture RBPs. We first defined relevant time points for identifying RNA partners. *B. subtilis* was grown in Difco sporulation medium (DSM) and observed at various times by phase-contrast microscopy (Fig. 3A). Sporulation, although not synchronized across cells, starts when nutrients are depleted. After 3 h in DSM, only vegetative cells are observable. The first forespores appear after 6–7 h in DSM. By 7 h, the population consists of both vegetative and sporulating cells. After 12 h, late forespores and first mature spores are present, their number increasing over time. Compared to the conditions used in reference 17, which employed the "resuspension method," we observed a slight delay in sporulation in our experiments, which followed the "exhaustion protocol" (Fig. 2 and 3A). Early stages of sporulation ($\sigma^E$, $\sigma^F$) are triggered between 4 and 5 h in Fig. 2, while our observation in DSM shows first forespores around 7 h. Altogether, these observations suggest that most RNA-mediated regulation of sporulation occurs between 7 and 12 h in DSM. We therefore chose to characterize the RBPome after 3, 7, and 12 h of growth in DSM. To this end, we performed OOPS, which has the advantage of requiring less starting material than other similar approaches. This approach combines *in vivo* UV crosslinking to stabilize RNA-protein interactions with acidic guanidine-phenol-chloroform extraction to enrich crosslinked protein-RNA complexes (Fig. 3B). The resulting fractions were analyzed by mass spectrometry (Fig. 3C). Since this approach had never been used in Gram-positive bacteria, which possess a cell wall composed of a thick layer of peptidoglycan, we adapted the protocol for *B. subtilis* and optimized the lysis conditions and UV irradiation (see Materials and Methods). To assess crosslinking and lysis efficiencies, we purified the interphase from cells subjected

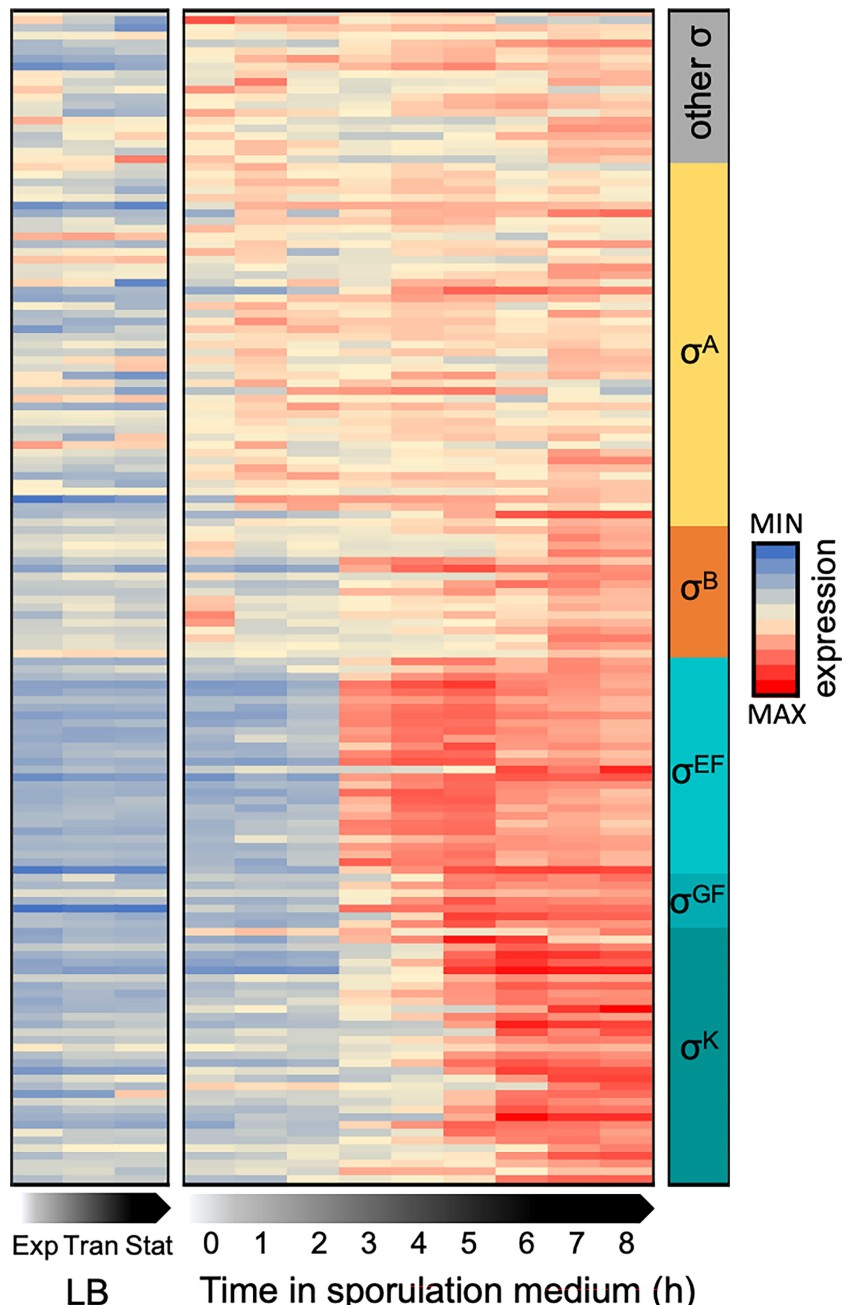

**FIG 2** Heatmap showing normalized expression of the 152 independently transcribed sRNAs in LB (exponential, transition, and stationary phases) and along sporulation (0 to 8 h), sorted by predicted or known dependency on σ factor(s) (shown in the right column) (17, 33). Expression at each time point is shown as a fraction of the total expression across all shown conditions (sum of normalized quantities across each line = 1). As expected, sRNAs dependent on $\sigma^E$, $\sigma^F$, $\sigma^G$, and $\sigma^K$ exhibit strongest expression during sporulation.

or not to UV and treated it with protease. RNAs were recovered intact from the interphase (Fig. 4A) and quantified (Fig. 4B). We observed that RNAs in the interphase of noncrosslinked samples represented less than 5% of total RNAs, consistent with efficient lysis (34). UV crosslinking increased protein-bound RNA recovery to ~20% of total RNAs (Fig. 4B). Notably, even without UV, RBPs were recovered in the interphase. Proteins isolated from the interphase ("RBPs") and from the organic phase (free proteins or "FP") were analyzed by mass spectrometry followed by label-free quantification.

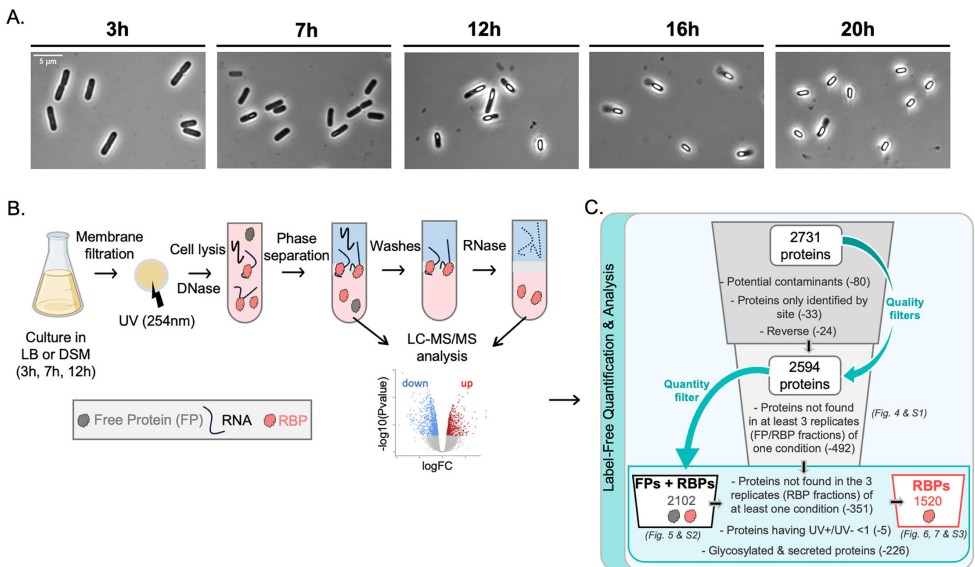

FIG 3 (A) Phase contrast microscopy images of *B. subtilis* cells after 3, 7, 12, 16, and 20 h growth in DSM. (B) Overview of OOPS protocol. To purify the whole RBPome involved in sporulation, we carried out OOPS at 3, 7, and 12 h in DSM. This approach combines *in vivo* UV crosslinking to stabilize RNA-protein interactions and acidic guanidine phenol chloroform extraction (AGPC, commonly known as TRIzol) to enrich crosslinked protein-RNA complexes. After culture, cells are rapidly filtered, subjected to UVC crosslinking, and lysed in the presence of DNases. Lysates are then phase separated using TRIzol. Free RNAs go in the upper aqueous phase, FPs in the lower organic phase, and crosslinked RNA-protein complexes sharing properties of both RNAs and proteins in the interphase. After isolation, the interphase is enriched in crosslinked RNA-RBP complexes by several rounds of separation (washes) and then treated with RNase and subjected twice to phase separation in order to recover only the RBPs. RBPs are then purified from the organic phase. Both FPs and RBPs are analyzed by liquid chromatography mass spectrometry (LC-MS) and quantified by label-free quantification (LFQ). (C) Filters applied for proteomic data analysis. Using MaxQuant Perseus software, 2,731 proteins were listed in all the samples analyzed. After utilization of quality filters (potential contaminants, proteins only identified by site, reverse), 2,594 proteins left. Then data were filtered to preserve only the proteins identified in at least three samples from one of the replicates (2,102 proteins left corresponding to both FPs and RBPs). Finally, RBPs were defined as the proteins once again identified in at least three replicates of a condition within the RBP fraction (1,751 proteins left). Then we filtered out proteins showing a UV+/UV− ratio lower than 1 (1,746 proteins left) and proteins annotated as secreted and glycoproteins that could be found in the interphase non-specifically (1,520 RBPs left).

To validate the approach, we calculated Pearson correlation coefficients to compare label-free quantification scores (LFQ) of proteins across the three biological replicates in different fractions (Fig. 4C and Fig. S1A, Table S2) and performed principal component analysis (PCA) (Fig. S1BC). The resulting correlation matrix showed the highest correlations for the RBPome (Fig. 4C) and the free proteome (Fig. S1A) recovered between cells grown 3 h in LB or DSM medium, where cells are still in vegetative state (Fig. 3A). However, the RBPome differed more after 7 or 12 h of growth in DSM, when sporulation has begun (Fig. 3A).

PCA confirmed high reproducibility of RBPome recovery between replicates in each condition (Fig. S1B). Replicates for the free proteome exhibited more variability, especially for samples grown for 7 h in DSM (Fig. S1C). As observed in the Pearson correlation matrix, PCA showed that biological triplicates of LB and 3 h DSM samples were closely grouped while data sets from 7 and 12 h DSM samples clustered independently (Fig. S1BC). Moreover, as expected, the RBP fraction exhibited low correlation with the FP fraction (Fig. S1A), consistent with specific isolation of RBPs in the interphase.

To assess the effect of UV, we represented the distribution of correlation coefficients between replicates within each condition as box plots (Fig. 4D). Although UV did not affect the FP recovery (Fig. S1A; Table S2), UV-crosslinked RBP fractions exhibited higher

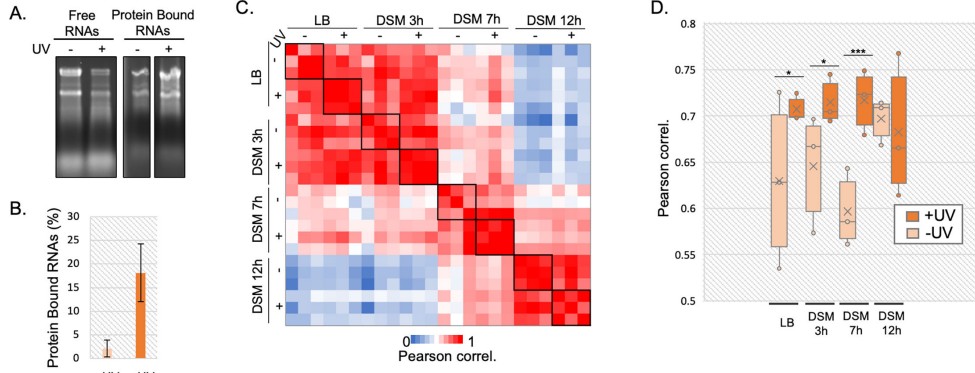

**FIG 4** (A) 1/20th of free RNAs and 1/3rd of protein-bound RNAs isolated from the aqueous phase and interphase, respectively, were separated on a 1.5% agarose gel and stained with ethidium bromide. (B) The proportion of isolated protein-bound RNAs with and without UV irradiation was calculated and displayed as a histogram. (C) Correlation matrix displayed as a heat map showing Pearson correlation between data sets (with LFQ intensities) obtained for each individual replicate of RBP fractions in LB and along sporulation process in DSM medium. (D) Distribution of Pearson correlation coefficient within triplicates for each growth condition. Higher coefficients indicate greater similarity between biological replicates.

correlations between replicates than non-irradiated samples except at 12 h in DSM (Fig. 4C and D). Indeed, as observed by microscopy (Fig. 3A), mature spores are present at 12 h. UV resistance being a characteristic of spores, they are likely less affected by crosslinking. This probably introduces variability among data sets. However, UV crosslinking allows recovery of greater quantities of RBPs with higher reproducibility across replicates (Fig. 4B).

## OOPS allows the specific purification of RBPs along sporulation

We then tested whether the recovery of a protein in the RBP fraction depended solely on its RNA-binding ability and not on its production level. We thus compared the distribution of relative abundance between the RBP and FP fractions. Proteins were sorted by relative abundance in the FP fraction, and their LFQ intensities were displayed as a heatmap (Fig. 5A and Fig. S2A, Table S3). We observed that LFQ intensity distributions were not correlated; some highly produced proteins exhibited low LFQ in the RBP fraction, while many low-abundance proteins were highly recovered in the RBP fraction. The recovery of a protein in the RBP fraction is thus independent of its production level.

Furthermore, to assess whether the RNA-binding profile during sporulation was dependent on protein production, we calculated, for each protein, the correlation coefficient between the recovery profiles in both fractions (see Supplemental Materials and Methods, Fig. 5B, Table S3). This coefficient indicates how differently a protein is recovered at each point between the two fractions. All fraction coefficients in our data ranged between 0.29 and 1 (median = 0.82) (Fig. S2B). For RBPs always bound to RNA, we expected a high correlation (close to 1). This is illustrated by two proteins that constitutively bind RNA: RpoE (Fig. 5C), a subunit of RNA polymerase, and RnpA, a protein component of RNase P, that have fraction correlation coefficients of 0.94 and 0.92, respectively (Table S3). In contrast, the proteins that bind RNA transiently have coefficients lower than the median value of 0.82 corresponding to the median value (Table S3). For example, this is the case of anti-terminators regulating operons by binding a terminator RNA structure only in the presence of an inducer (35) (Fig. 4D), like HutP, with a low fraction correlation coefficient of 0.69, LicT (0.77) (36), SacT (0.71) (37), and RplT (0.78) (38). Notably, we observed that these fraction correlation coefficients were independent of RBP abundance (Fig. 5A, B, and Fig. S2C).

The top 500 proteins identified in the RBP fractions (from all samples combined) were analyzed for Gene Ontology (GO) terms enrichment. Notably, one protein can be

found in several GO terms (Fig. 5E). Among the 10 most enriched GO terms, seven are associated with RNA function. Remarkably, the three remaining enriched GO terms are related to metabolic enzymes, consistent with previous observations (13, 29–31). This suggests that these enzymes may bind RNA in response to nutrient availability to regulate gene expression. Notably, the well-known moonlighting protein CitB, both a metabolic enzyme and RBP under iron-starvation conditions (39), whose deletion results in a strong sporulation defect (40), was found in the top 100 RBPs isolated after growth in LB and DSM (3, 7, and 12 h).

## The RBPome is extensively remodeled during the sporulation process

Among the 2,102 proteins identified in both the FP and RBP fractions, 1,520 passed the filters to be considered as putative RBPs (Fig. 3C): they were identified in at least three biological replicates of one condition in the RBP fractions, exhibited enrichment in crosslinked samples compared to non-crosslinked samples, and were to be secreted or to bind or interact with glycans (including spore crust proteins). Indeed, glycoproteins may not be specifically enriched in the interphase (32, 41).

We then examined how specific these data sets were to each condition (Fig. 6; Table S4). A total of 70% of proteins (1,049/1,520) were recovered in RBP fractions across all four conditions; among them were almost all (54/60) ribosomal proteins encoded in the genome. Only 2% of proteins seemed to be exclusive to the vegetative state (LB and or DSM 3 h), while more proteins were recovered specifically during the sporulation process. Indeed, more than 13% of RBPs appeared specific to DSM 7 and 12 h (either exclusively or at both time points). This indicated that RBPs exhibited distinct RNA-binding and production profiles throughout sporulation.

## Classification of RBPome in different clusters and identification of known RBPs

To compare production and RNA-binding ability of RBPs and group together proteins exhibiting similar behavior, we performed clustering analysis (Euclidean distance-based) on normalized intensities, where recovery score at each condition is the fraction of the total recovery of this protein in a given fraction (Fig. 7, Table S5). Each cluster was annotated for known RBPs, essential genes, and known or predicted function. GO-term enrichments were also assessed for each cluster. The distribution of unnormalized LFQ values per fraction is shown in Fig. S3.

Clusters 1, 2, and 3 contain proteins specific to vegetative growth. Proteins of cluster 1 exhibit exclusive high recovery in LB in both fractions. Clusters 2 and 3 include abundant proteins peaking at 3 h in DSM but with maximal RNA-binding in LB, with decreasing abundance over time in DSM, consistent with reduced metabolic activity and repression of transcription and translation during sporulation (42). The main distinction between clusters 2 and 3 is the weaker correlation between protein production and RNA-binding profiles in cluster 3. Enrichment was observed for pathways including respiration, transcription, metabolism, cell division, and chemotaxis. YvcJ, a homolog of *E. coli* sRNA-binding RNase adaptor RapZ (43, 44), SpoVG, potentially involved in sRNA processing (16), 12 RNases and RNase cofactors, and cold shock proteins CspB, CspC, and CspD, preventing the formation of translation-inhibiting mRNA structures under stress (45, 46), were all recovered in clusters 2 and 3 (Table S6). Cluster 4 contains abundant proteins (Fig. S3), enriched in stress-response and metabolic proteins, highly produced at 3 h in DSM but binding RNA primarily at 12 h, suggesting stress-induced RNA interaction. Of 231 proteins, 32 are known RBPs, including RNA chaperones Hfq and Jag, not previously identified as an RBP in *B. subtilis* but known to bind sRNAs in other bacteria (47–49). As for cluster 4, proteins of cluster 5 exhibit poor correlation between production (peaking at 3h and 12 h in DSM) and RNA-binding (detectable only at 12 h), suggesting transient RNA interactions. While no GO terms were enriched, many metabolic enzymes were present. Low abundance may contribute to noisy recovery.

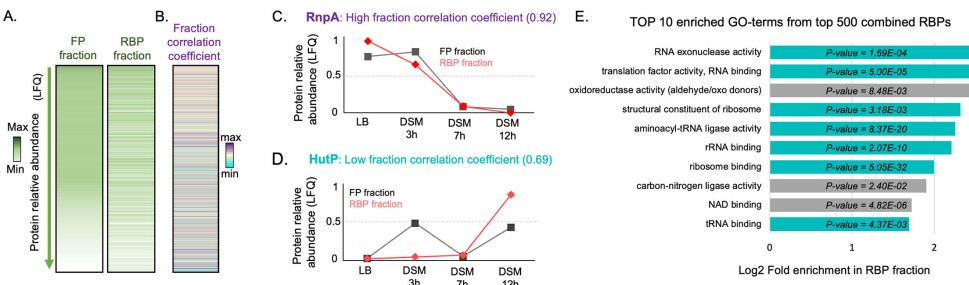

**FIG 5** (A) Heatmap showing protein abundance (LFQ) in both FP and RBP fractions. Within each fraction, LFQ intensities from all four conditions are summed for each protein. (B) Corresponding fraction correlation coefficient. The heatmap is sorted by average abundance in FP fraction. A high fraction correlation coefficient corresponds to proteins always bound to RNA, and their abundance profiles in both fractions are similar. A low fraction correlation coefficient corresponds to proteins transiently bound to RNA; they exhibit different profiles in both fractions. All values are displayed in Table S3. Representation of normalized abundance profiles in both FP and RBP fractions for RnpA, a protein always bound to RNA, exhibiting a high dynamic correlation coefficient (C), and HutP, a protein bound to RNA only in certain conditions (D). (E) Top 10 enriched GO terms from the top 500 RBPs from the different conditions (all *P*-values < 0.025).

Cluster 6 is enriched in rRNA and amino acid metabolism enzymes. Despite modest production, these proteins display strong RNA-binding profiles (Fig. S3), likely reflecting constitutive RNA-binding to support survival (50). Cluster 7 includes proteins detected only at 3 h in DSM, with RNA-binding closely mirroring their production profiles. Low abundance in the RBP fraction suggests only a subset of the produced proteins engages in RNA-binding (Fig. S3).

RBPs found in clusters 8, 9, and 10 were highly enriched in sporulation proteins and displayed similar profiles between both fractions (cluster 8: 29/58 RBPs; cluster 9: 48/122; cluster 10: 51/106). Cluster 8 contains RBPs mainly produced after 7 h in DSM, while those in cluster 9 are found at both 7 and 12 h. Although relatively low, their abundances are higher in the RBP than in the FP fraction (Fig. S3), suggesting that a significant proportion is RNA-bound. Notably, factors in the fatty acid metabolism pathway are highly enriched, consistent with the increase of lipid production required for spore formation (51, 52). Enrichment was also observed for amino acid metabolism proteins (clusters 6, 8, 9). In cluster 10, RBPs are specifically produced after 12 h in DSM, when most cells are metabolically inactive. Accordingly, this cluster is depleted in proteins related to gene expression, translation, RNA metabolism, and other cellular processes.

Our data sets confirmed the robustness of the experimental approach with over 230 known RNA-associated factors identified in the RBP fractions, with additional proteins characterized or suggested in other bacteria as sRNA-interacting factors such as YvcJ, Hfq, SpoVG, Kre (YkyB, cluster 1) (53), Jag, and KhpA (cluster 8) (47–49). This confirmed that our experimental conditions allowed the identification of RBPs. We also recovered 66 of the 116 Sm-domain and like-Sm containing proteins encoded by the *B. subtilis* genome (Table S10), such as Hfq. These proteins play fundamental roles in RNA processing and gene expression regulation across domains of life (54).

Among the 30 known RNases and RNase cofactors in *B. subtilis*, 25 were identified using OOPS, suggesting process- and target-specific functions (Table S6). Sporulation-specific exoribonucleases KapD and nano-RNase B (NrnB) were identified in clusters 6 and 9, respectively, displaying peak production and RNA-binding activity during late sporulation. This matched their known roles in regulating sigK mRNA stability in the mother cell (KapD) and in mediating terminal mRNA breakdown in the forespore (NrnB) (55, 56). RNases PH and R, although not sporulation-specific, were also highly abundant during this process. Beyond RNases, three of the four RNA helicases encoded in the *B. subtilis* genome, CshA, CshB, and CshC (YfmL), were identified in clusters 3, 2, and 6, respectively. These proteins, conserved among Bacilli, have known or suspected roles in RNA turnover (57), ribosome assembly, sporulation, and stress adaptation (58, 59).

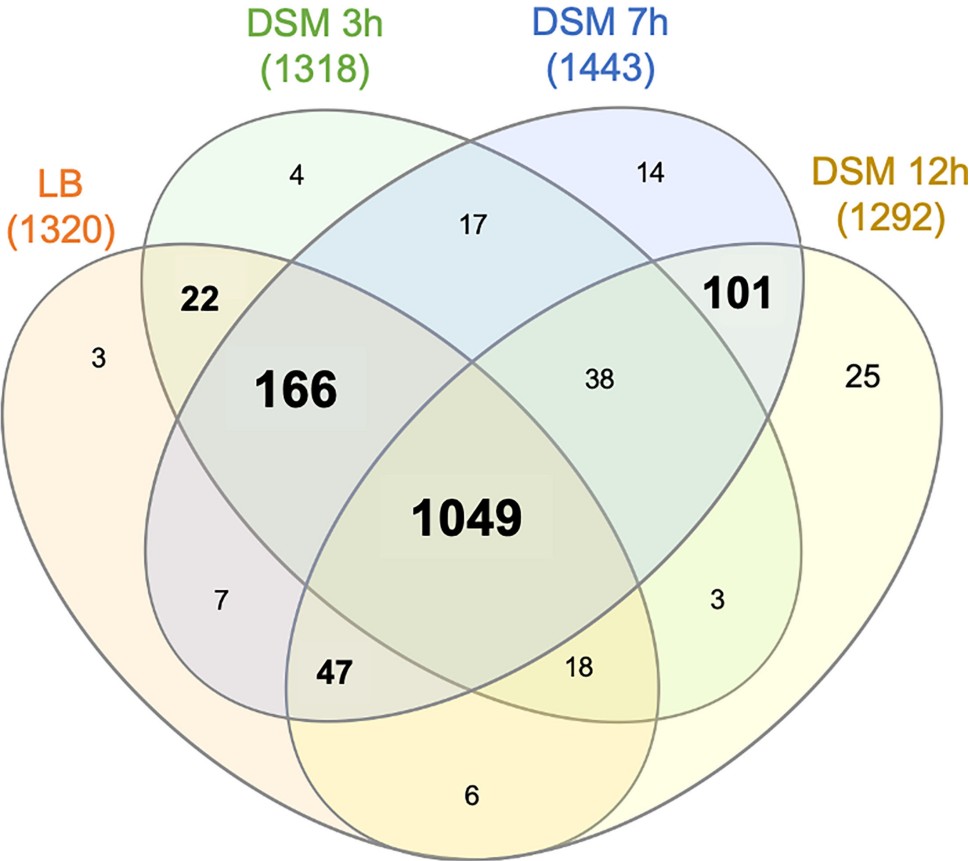

**FIG 6** Venn diagram showing the overlaps between RBPs identified in cells grown in LB and after 3, 7, and 12 h in DSM.

## Highlight on unknown proteins binding RNA specifically during sporulation

Among OOPS-identified RBPs, we expect sRNA partners involved in setting up sporulation to harbor RNA-binding abilities that fluctuate along sporulation. We consequently focused on RBPs whose presence significantly increased at 7 h in DSM, conditions where more sRNAs are expressed and most cells are undergoing sporulation but are still not mature spores. We decided to focus on clusters 8 and 9 (Table S7), consisting of 180 proteins, including 83 proteins known as involved in sporulation.

As noted in the global results, a significant portion of proteins in clusters 8 and 9 are linked to cell metabolism, suggesting they are possibly moonlighting enzymes, with two or more physiologically relevant biochemical or biophysical functions (60, 61). Among the 46 proteins (out of 180) with uncharacterized functions, 39 are annotated as putative enzymes, while the remaining 6, listed in Table 1, have unknown function. We focused on these to explore their potential ncRNA partners, examining conservation, structural homology, σ factor dependency, and available data from literature and *Subtiwiki* database (10). These different proteins exhibit diverse recovery scores (FP fraction) and enrichment in the RBP fraction. Except for YxnB, known to stabilize its own operon mRNA involved in adaptation to nutrient starvation and transition to sporulation (62), all are conserved in other spore-forming Bacillota, supporting a possible role in sporulation. YqxD contains a PIN-like domain associated with ribonuclease activity (63) and has been shown to affect sporulation initiation (64). YxkF and YdeE are putative transcriptional regulators with Helix-Turn-Helix (HTH) domains, both accumulating during sporulation. YjzH is overproduced under glucose exhaustion (17) and has strong structural similarity to SpeG (65), a conserved polyamine acetyltransferase involved in *Bacilli* biofilm formation and reported to regulate an *E. coli* sRNA (66). Interestingly, these six proteins include KhpA (YlqC), a highly conserved putative RNA chaperone, uncharacterized in *B.*

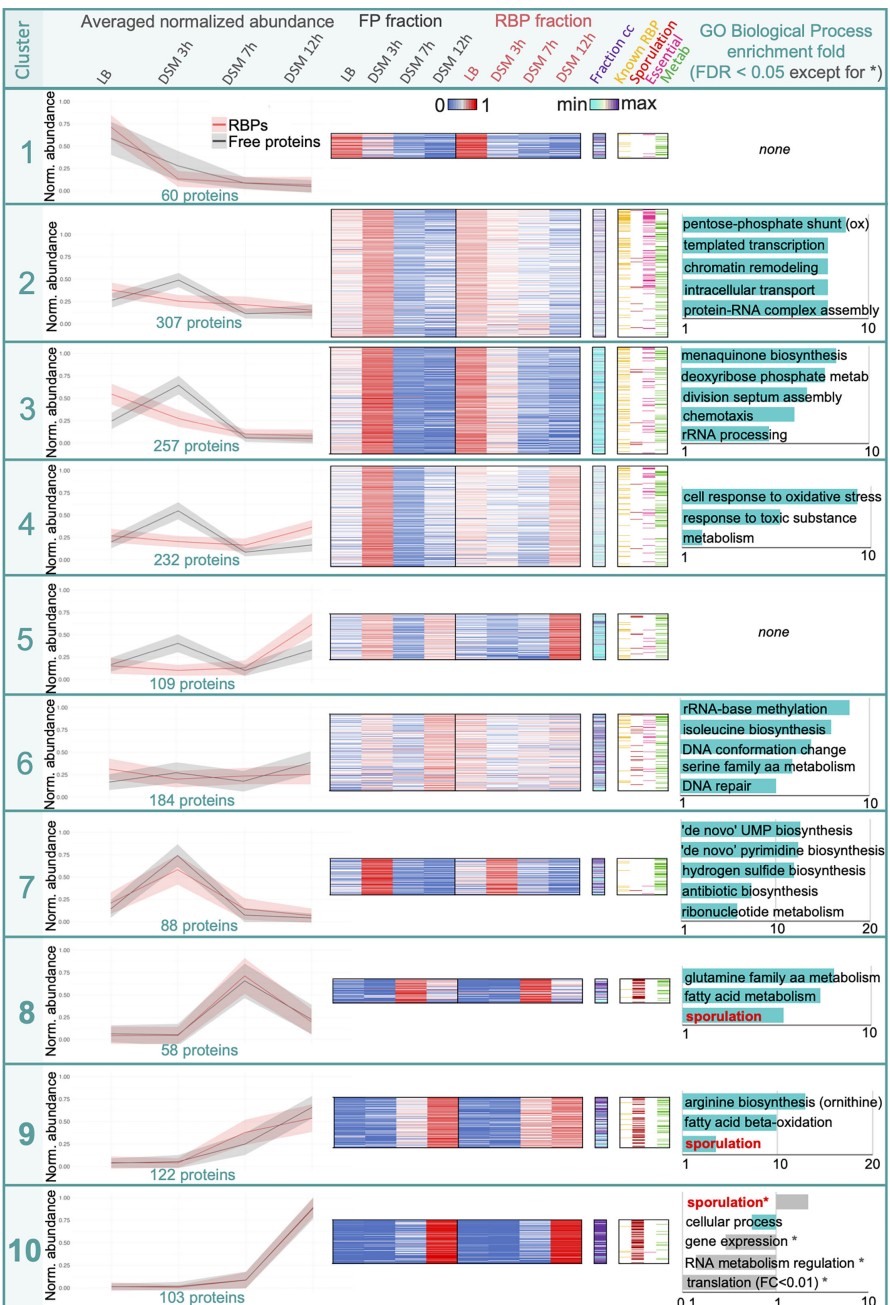

**FIG 7** Normalized LFQ intensities in both FP and RBP fractions (corresponding to production and RNA-binding profiles, respectively) displayed as a heatmap. Each line corresponds to a protein. For each condition, abundance is expressed as a fraction of the total recovery of this protein across all four tested conditions, either in the FP or RBP fraction (sum of normalized quantity across each line = 1). Proteins have been clustered by Euclidean distance into 10 distinct groups according to their profiles. All values are displayed in Table S5.

*subtilis*. Altogether, our results show that these proteins bind RNAs during sporulation and thus may contribute to RNA-mediated regulation of this process.

## KhpA, an RBP with potential role in sporulation regulation

We further investigated the role of KhpA in sporulation. Among the sporulation-specific clusters (8, 9, and 10), KhpA exhibited the lowest fraction correlation coefficient (0.43), suggesting that it does not always bind RNA. KhpA is highly abundant in vegetative

**TABLE 1** Proteins of unknown function recovered in clusters 7 and 8[a]

| Gene | Cluster | Fraction correlation coefficient | LFQ in RBP fraction DSM 7 h | Molecular weight (kDa) | σ factor | Notes | Conservation |
|------|---------|----------------------------------|------------------------------|------------------------|----------|-------|--------------|
| ylqC | 7 | 0.43 | 0.8E + 07 | 8.9 | σ$^A$ | Putative RNA chaperone KhpA, KH domain, heterodimerizes with KhpB in other bacteria | + + Conserved |
| yxnB | 7 | 0.83 | 1.1E + 07 | 18.9 | σ$^A$ | Operon important for transition to sporulation, mRNA stabilizer | Unique |
| yqxD | 8 | 0.76 | 0.7E + 07 | 22.4 | σ$^{AH}$ | Strongly expressed, in operon with sigA, DUF188 | Bacillota (Bacillus, Streptococcus) and proteobacteria |
| yxkF | 8 | 0.73 | 1E + 07 | 34.7 | σ$^A$ | Putative transcriptional regulator, repressed by glucose | Bacillota (Bacillus, Streptococcus) and some actinobacteria |
| ydeE | 8 | 0.78 | 1.6E + 07 | 7.2 | | HTH domain, putative transcription factor | + + Conserved |
| yjzH | 8 | 0.85 | 3.9E + 07 | 7.7 | σ$^W$ | DUF4177, structural homology with SpeG | No sequence conservation but structural homology within Bacillota and E. coli |

[a]No known or predicted σ factor.

cells, and its production decreases during sporulation (Fig. 8A). However, it is detected in the RBP fraction only after 7 h in DSM, and exclusively in the UV-irradiated sample. This suggests that KhpA specifically binds RNA during early sporulation, despite being less expressed than in vegetative cells. To explore KhpA RNA-binding activity, we constructed a strain expressing KhpA fused to an HF (6× histidines, 1× FLAG domain) tag under the control of its native promoter. After growth in LB to the transition phase and DSM for 7 h, cells were subjected to UV crosslinking. KhpA-HF was then tandem-affinity purified under both native and denaturing conditions, ensuring high stringency in the recovery of authentic RNA-protein interaction sites. Following RNA-specific radioactive labeling (see Materials and Methods) and gel separation, KhpA-associated RNAs were detected (Fig. 8B), under sporulation conditions and, to a lesser extent, during vegetative growth, although no signal was detected in OOPS.

The majority of the KhpA sequence consists of a KH domain, a well-characterized RNA-binding motif, highly conserved among bacteria (Fig. 8C and D). KhpA has been previously reported to interact with RNAs, including sRNA, in *Streptococcus pneumoniae* or *Fusobacterium nucleatum* (47–49) and was shown to be able to form homodimers in *S. pneumoniae* (68). Using AlphaFold 3 (67), we noticed that interaction with RNA may enhance the stability of the dimeric form of KhpA. Indeed, without RNA, the interference predicted template modeling (ipTM) score was 0.35, and in the presence of a poly-U RNA, ipTM score increased to 0.81. To assess KhpA's potential role in sporulation, we analyzed a strain lacking *khpA*. The deletion strain exhibited a slight but reproducible increase in sporulation efficiency, complemented by expression of *khpA* under the control of its native promoter, suggesting a weak regulatory role in this process (Fig. 8E).

## SpoVR, a protein affecting sporulation with potential RNA-mediated regulation

Within cluster 8, SpoVR, a protein whose absence reduces spore resistance to heat (69), caught our attention. It is conserved in many spore-forming bacteria, from *bacilli* to Gram-negative *Myxococcales*, and participates in spore cortex formation in *B. subtilis* (69)

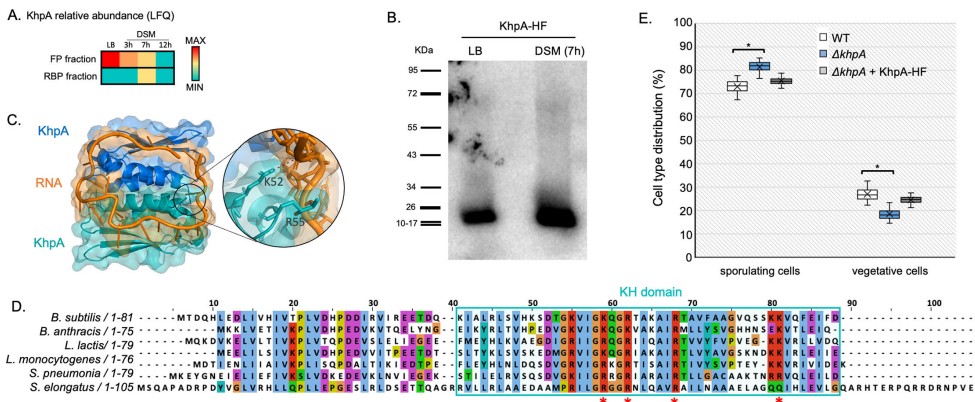

**FIG 8** (A) KhpA relative abundance (LFQ) in OOPS FP and RBP fractions, along sporulation. (B) *In vivo* RNA-binding assay on KhpA-HF in LB (transition phase) and DSM (7 h). $^{32}$P-labeled RNAs crosslinked to KhpA-HF were visualized using a PhosphorImager. (C) Structure of KhpA homodimer in complex with a 20-U RNA, predicted by AlphaFold 3 (67). (D) Sequence and domain alignments of KhpA from different bacteria (*B. subtilis, B. anthracis, Lactococcus lactis, Listeria monocytogenes, Streptococcus pneumoniae, Synechococcus elongatus*). KH domain is highlighted in blue; its minimal structure is characterized by the presence of two α-helices flanking a β-sheet on each side, as well as a GxxG motif between the two α-helices. This motif, along with the amino acids of the adjacent β-sheet, forms an RNA-binding surface. Residues predicted to form the strongest bonds (K52, R55, R61, K74) with RNA are shown with a red asterisk. (E) Sporulation efficiency assay; fraction of sporulating cells (spores + forespores) is calculated relative to total cells (vegetative + forespores + spores) after 24 h in DSM. Tukey tests were performed and showed significant differences between WT and *ΔkhpA* strains (* = *P*-values <0.05). Differences between *ΔkhpA* and complemented strains exhibit *P*-values <0.01 for both cell types.

and *Myxococcus xanthus* (70). SpoVR was recovered specifically at 7 and 12 h in DSM, in both FP and RBP fractions (with approximately sevenfold higher recovery in UV-irradiated samples), and exhibited the highest RNA-binding ability after 7 h in DSM (Fig. 9A). To confirm this RNA-binding ability, an *in vivo* RNA-binding assay was performed as previously done for KhpA. A strain expressing SpoVR-HF under the control of its σ$^E$-dependent promoter was analyzed during vegetative growth (LB, transition phase) and sporulation (7 h DSM). Radioactive RNAs crosslinked to SpoVR-HF were detected as expected during sporulation (Fig. 9B). Interestingly, AlphaFold 3 predicted that a poly-U RNA binds SpoVR within a positively charged surface cavity (ipTM = 0.62), despite the absence of a known RNA recognition domain (Fig. 9C).

## DISCUSSION

In this paper, we followed the evolution of the RNA-binding proteome during sporulation. By applying OOPS for the first time to a Gram-positive bacterium, we obtained a comprehensive overview of the RBPome during both vegetative growth and sporulation in *B. subtilis*. Interestingly, our results show that numerous sporulation factors engage in RNA-binding, such as GerW and YutG, identified in cluster 10. This reinforces the hypothesis that the stable RNAs present in mature spores (71, 72), and the large number of sRNAs controlled by σ$^G$ and σ$^K$ (Fig. 2) interact with RBPs to regulate sporulation. However, technical challenges remain in studying these complexes, particularly in lysing spores and overcoming the resistance of mature spores to UV irradiation. Addressing these limitations will be essential for further investigations.

Technological advances in studying RNA-protein interactions have revealed a surprisingly high number of unconventional RBPs (73). Many of these proteins, such as metabolic enzymes (74, 75), were previously thought to be unrelated to RNA metabolism. While often overlooked, multifunctionality in proteins is common: more than 500 proteins have been classified as moonlighting proteins. Among these, the aconitase CitB (39), which interacts with RNA, was recovered among the top 300 most abundant RBPs. Beyond enzymes, 57 HTH domain-containing proteins (out of the 245 encoded in the

A. SpoVR relative abundance (LFQ)

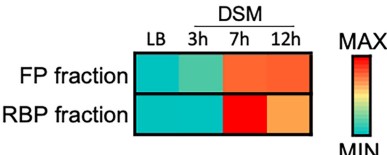

B.

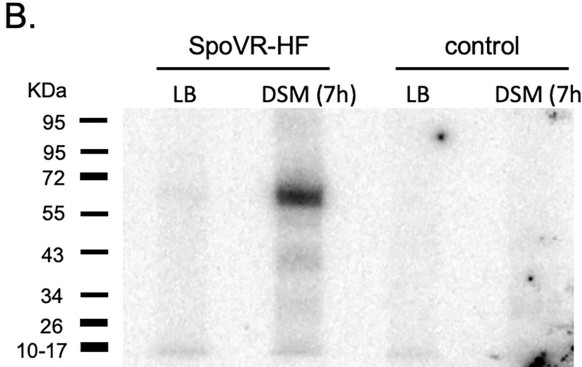

C.

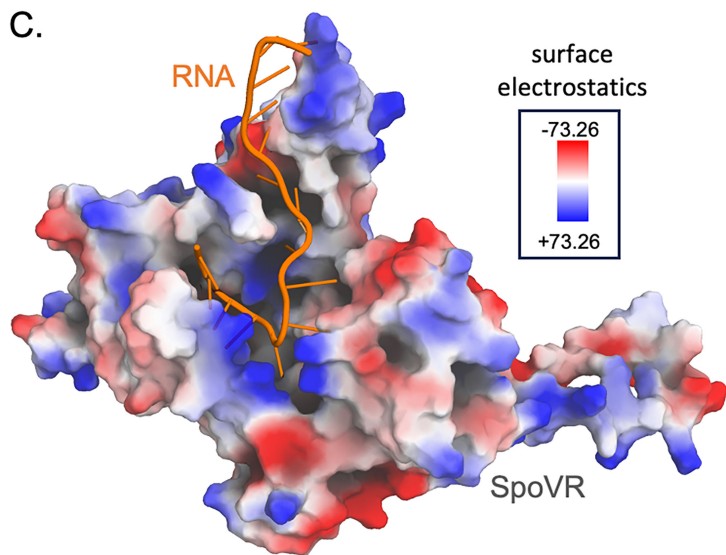

**FIG 9** (A) SpoVR relative abundance (LFQ) in OOPS FP and RBP fractions, along sporulation. (B) *In vivo* RNA-binding assay on SpoVR-HF in LB (transition phase) and DSM (7 h). $^{32}$P-labeled RNAs crosslinked to SpoVR-HF were visualized using a PhosphorImager. (C) Structure of SpoVR in complex with a 15-U RNA, predicted by AlphaFold 3 (67). Electrostatic surface charges are represented as a color gradient ranging from blue (positive) to red (negative).

genome) were identified across the different conditions (Table S10). Usually considered transcriptional regulators that bind DNA, our results suggest that some of them can also bind RNA under specific conditions. This has already been shown for CcpA, the master regulator of catabolite repression in several Gram-positive bacteria, which can bind hundreds of RNAs and may regulate their stability (30, 76). Structural analyses have recently shown how an HTH domain can specifically discriminate between DNA and RNA (76).

Regarding KhpA, it has previously been reported to interact with RNAs, including sRNAs, in complex with the protein KhpB in *Clostridioides difficile, S. pneumoniae,* or *F. nucleatum* (47–49). This raises the question of whether KhpA recognizes conserved RNA targets across species, particularly in spore formers such as *C. difficile*. KhpB (also named Jag) was found among OOPS-isolated RBPs in cluster 4. Both KhpA and Jag contain RNA-binding domains and have conserved homologs in Gram-positive bacteria (77). Most research about KhpA and KhpB has been done in *S. pneumoniae,* where KhpA forms homodimers or heterodimers with KhpB. Strains lacking either protein phenocopy each other (68, 78), indicating joint roles in growth and cell elongation. Our data suggest that KhpA may have a specific role in *B. subtilis* sporulation. Whether this role is exclusive to KhpA or shared with KhpB remains to be explored.

To conclude, the identification of novel RBPs will serve as a stepping stone for many mechanistic studies. While few studies have attempted to identify and characterize global RNA chaperones in Gram-positive bacteria using *in vivo* global approaches, post-transcriptional regulation remains much less understood than in Gram-negative bacteria. Thus, it is necessary to further explore these potential new RBPs and their RNA targets. Characterizing RBPs that regulate RNA-mediated bacterial adaptation will enhance our understanding of the physiology and the resilience of *B. subtilis* and Gram-positive bacteria in general.

## MATERIAL AND METHODS

### Bacterial strains, plasmids, media, and genetic manipulations

All *B. subtilis* strains are derived from wild-type 168 and listed in Table S8. Fragments encompassing the *khpA* gene and its own promoter were amplified from genomic DNA, assembled with *Gibson Assembly Master Mix* (NEW ENGLAND BioLabs), and cloned within the pAC5 vector (79). An HF tag (HHHHHHRPDYKDDDDK) was added to the C-terminal end of *khpA* by adding the sequence in the reverse primer used for PCR amplification. The *khpA* deletion strain was transformed with the pAC5-Khpa-HF plasmid to generate the complementation strain by insertion at the *amyE* locus. Strains were grown either in LB medium or in DSM supplemented with KCl 0.1%, $MgSO_4$ 1 mM, NaOH 0.5 mM, $Ca(NO_3)_2$ 1 mM, $MnCl_2$ 0.01 mM, and $FeSO_4$ 0.001 mM.

### Orthogonal organic phase separation

OOPS was carried out at defined time points in DSM (3, 7, and 12 h). This approach combines *in vivo* UV crosslinking to stabilize RNA-protein interactions and acidic guanidine phenol chloroform extraction (AGPC, known as TRIzol) to enrich crosslinked protein-RNA complexes. After UV irradiation, the bacterial cells were lysed. The protocol was optimized for *B. subtilis,* whose resistant cell wall is mainly composed of peptidoglycan. Next, biphasic protein extraction was performed, followed by purification of RBPs. The detailed protocol is described in Supplemental material.

### Proteomic sample preparation

All RBP samples or 1/10th of the FP samples were denatured in NuPage 1× loading buffer for 5 min at 65℃, then separated on a NuPage Bolt 8% Bis-Tris Plus gel for 4 min at 150 V (the samples migrated as a thin band in the stacking part of the gel). Gels were stained with Coomassie Blue or Imperial Blue for 1 h, destained for 3 × 15 min, then left overnight before each strip was cut and placed in a low-binding tube. Gel slices were reduced with dithiothreitol, alkylated with iodoacetamide, and digested by Trypsin/LysC Mix (Promega) following the protocol previously described (80). Samples were analyzed using a hybrid Q-Orbitrap mass spectrometer (Q-Exactive Plus, Thermo Fisher Scientific, USA) coupled to a Vanquish Neo UHPLC system (Thermo Fisher Scientific, USA). A detailed protocol for MS experiments and analysis is provided in Supplemental material.

## Sporulation efficiency assay

Cultures were inoculated at $OD_{600nm}$ = 0.1 from DSM precultures of each strain and grown in DSM medium for 24 h at 37°C with shaking at 170 rpm. Sporulation efficiency was assessed by phase-contrast optical microscopy, counting the number of spores, forespores, and vegetative cells in 24 h DSM cultures using images acquired with a Zeiss Upright Axio Imager M2 microscope. Cell counts (total >1,000) were performed across 10 to 20 microscopy fields per experiment. This approach was repeated for three biological replicates. Statistically significant differences between strains were determined by ANOVA using Tukey's test.

## *In vivo* RNA-binding assay

RNA-binding assays were performed on strains expressing either KhpA or SpoVR fused to an HF tag, as well as a WT strain as control, grown in LB to the transition phase or in DSM for 7 h. UV-crosslinking and lysis in the presence of RQ1 DNase (Promega) were performed as described for OOPS. RNA-protein complexes were purified on Magnetic Anti-FLAG M2 beads (Merck). As described in reference 81, RNAs were partially digested with RNase A/T1, leaving only the protein "footprint." Trimmed complexes were denatured using 6M guanidinium, immobilized on Ni-NTA affinity resin, and washed under denaturing conditions to remove co-purifying proteins and complexes. Following RNA 5' labeling with [32]P using nucleic acid-specific T4 PNK (NEB), RNA-protein complexes were eluted and separated on a denaturing SDS-PAGE (NuPAGE Bis-Tris, Invitrogen), transferred to a nitrocellulose membrane, and exposed to autoradiography using an FLA-5100 Typhoon variable-mode Imager (Amersham Biosciences).

## ACKNOWLEDGMENTS

We thank the IM2B for financial support (grant for newcomers). We thank Regine Lebrun from the Proteomics Platform from the Institut de Microbiologie de la Méditerranée (CNRS, FR3479), which is part of the proteomic network "Marseille Protéomique" (MaP) IBISA and Aix-Marseille Université-labeled, for her expertise. We also thank Jörg Vogel for critical reviewing of the manuscript and Nicolas Waisbord for helpful discussion about correlation analysis.

## AUTHOR AFFILIATIONS

[1]Laboratoire de Chimie Bactérienne, UMR7283, IMM, Aix-Marseille Université - CNRS, Marseille, France
[2]IMM Proteomics platform, Marseille-Proteomique (MaP), Marseille, France

## AUTHOR ORCIDs

Clémentine Delan-Forino ⓘ http://orcid.org/0000-0003-0915-1211

## FUNDING

| Funder | Grant(s) | Author(s) |
| --- | --- | --- |
| IM2B Starting Grant | | Clementine Delan-Forino |

## AUTHOR CONTRIBUTIONS

Thomas Kaboré, Data curation, Formal analysis, Investigation, Methodology, Software, Validation, Writing – original draft | Maya Belghazi, Data curation, Formal analysis, Methodology, Software, Validation | Christophe Verthuy, Data curation, Investigation, Methodology | Anne Galinier, Funding acquisition, Project administration, Supervision, Writing – original draft, Writing – review and editing | Clémentine Delan-Forino, Conceptualization, Data curation, Formal analysis, Funding acquisition, Investigation, Methodology, Project administration, Resources, Software, Supervision, Validation, Visualization, Writing – original draft, Writing – review and editing

## DATA AVAILABILITY

The mass spectrometry proteomics data have been deposited to the ProteomeXchange consortium via the Pride Proteomics Identification Database partner repository with the data set identifier PXD061929.

## ADDITIONAL FILES

The following material is available online.

### Supplemental Material

**Fig. S1 (mSystems00496-25-S0001.tiff).** Correlation matrix and principal component analysis.
**Fig. S2 (mSystems00496-25-S0002.tiff).** LFQ intensities.
**Fig. S3 (mSystems00496-25-S0003.tiff).** Distributions of LFQ intensities of proteins within each cluster.
**Supplemental material (mSystems00496-25-S0004.docx).** Text S1 and supplemental figure and table legends.
**Supplemental tables (mSystems00496-25-S0005.xlsx).** Tables S1 to S10.

### Open Peer Review

**PEER REVIEW HISTORY (review-history.pdf).** An accounting of the reviewer comments and feedback.

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
