## [Reviewer comments · mSystems]

Remodeling of RNA-Binding Proteome and RNA-mediated regulation as a new layer of control of sporulation

Thomas Kabore, Maya Belghazi, Christophe Verthuy, Anne Galinier, and Clementine Delan-Forino

Corresponding Author(s): Clementine Delan-Forino, Laboratoire de Chimie Bacterienne, UMR7283, AMU-CNRS, Marseille

Review Timeline:

Submission Date:	April 15, 2025
Editorial Decision:	May 9, 2025
Revision Received:	June 24, 2025
Editorial Decision:	July 2, 2025
Revision Received:	July 7, 2025
Accepted:	July 10, 2025

Editor: Sarah Svensson

Reviewer(s): The reviewers have opted to remain anonymous.

Transaction Report:

DOI: <https://doi.org/10.1128/msystems.00496-25>

Re: mSystems00496-25 (Remodeling of RNA-Binding Proteome and RNA-mediated regulation as a new layer of control of sporulation)

Dear Dr. Clementine Delan-Forino:

I am happy to report that both reviewers were positive about your manuscript, but both require some things be addressed prior to acceptance.

In particular, the manuscript requires extensive proofreading for grammar and conciseness. It might be useful to also provide line numbers to help with the review process. Please let me know if you need assistance with editing.

I look forward to reading a revised version!

Revision Guidelines

Sincerely,
Sarah Svensson
Editor
mSystems

Reviewer #1 (Comments for the Author):

Kabore et al describe their work on the RBPome of *Bacillus subtilis* and identify new RNA-binding proteins (RBPs). The authors have used OOPS to capture a time course of the RBPome during sporulation in an effort to identify RBPs required for sporulation. They characterise KhpA and SpoVR, and demonstrate that the former has a negative impact on the number of spore forming cells.

The data in the manuscript are valuable and will provide a useful resource for the community, but the presentation requires some work to make it accessible. The manuscript needs proofreading throughout. Some sections (eg: analysis of clusters) are overly verbose and distract from the key messages. Shortening the manuscript to focus on the key findings will improve readability.

Major comments:

1. Pg 6 second paragraph. The first RBPome for a Gram positive was PMID: 35610211, which is referenced later in the manuscript. I think the author mean the first time OOPS has been applied to a Gram positive.
2. Pg 11-14: 'Classification of RBPome in different clusters'. The authors present a very detailed description of each cluster, which seems unwarranted for clusters 1-7 and becomes an extended list of observations. Clusters 8 and 9 contain sporulation genes and the authors choose KhpA and SpoVR from these clusters. I suggest significantly condensing or reducing clusters 1-6.
3. Pg 14: "Highlight on unknown proteins ..." The authors focus on 7 unknown proteins from clusters 8 and 9. Again, each protein is described in a separate paragraph but there are no conclusions drawn from the list of observations. The authors conclude with "...our results show that these proteins bind RNA during sporulation process..." but it is not clear what results have been presented.
4. Pg 16: The data from Abrahamson et al is described in a few sentences and their structure is presented in Figure 8. It might be better to describe the main finding succinctly - KhpA is predicted to require RNA binding for stability - or repeat these analysis and AF3 structure predictions, which would also allow modelling with *B. subtilis* sRNA rather than poly(U).
5. Pg 17: khpA is shown to contribute to sporulation. The authors conclude on pg 20 that KhpB (Jag) does not have a specific role in sporulation, but it is not clear what supports this conclusion. Does a khpA jag double deletion have a more pronounced sporulation defect?
6. Based on work in *Streptococcus* and *Fusobacterium*, is there any indication of the likely RNA ligands for KhpA in *B. subtilis*? KhpB has been described in *Clostridioides difficile* (PMID: 37223250) that is also sporulating and may provide some clues to the likely ligands.
7. Pg 17: From the AF3 SpoVR-RNA structure, can the authors infer anything about how the RNA is binding to SpoVR?

Minor comments:

1. The manuscript needs careful proofreading for grammatical issues throughout. I have not listed them here as there are too many.
2. The phylum Firmicutes is now Bacillota (PMID: 34694987).

Reviewer #2 (Comments for the Author):

The paper by Kaboré et al describes the large-scale identification of RNA binding proteins in *B. subtilis*, a study that is long-overdue. Using OOPS (orthogonal organic phase separation) the authors identify over a thousand potential RNA binding proteins in both normal growth and sporulation conditions. The paper is well-written, in particular the introduction which sets the paper up well. I have only a few minor comments to improve the presentation of the manuscript.

I felt that a major section of the discussion (from bottom of P17 to top of P19) would have fit better in the results after the presentation of the 10 clusters. A resumé of the highlights of what was actually found will be expected earlier by the reader. It will shorten the discussion, but that's OK

Please use line-numbers to help review process!

P4 2 lines from bottom. To date, most research has focused

P6 and Fig 2. Please specify whether all 150 potential sRNAs are included in the heat-map

P7. The sentence starting "Those putative sRNAs ..." is long and awkward. Please rephrase

P7 middle. I suggest modifying the first sentence of the new section to 'To identify RNA-protein complexes involved in RNA-mediated regulation during sporulation, ...' to better make the connection with the previous section

P7 Please specify that the sporulation method used in this paper is the 'exhaustion' method, for sporulation experts. It could also be mentioned that the Nicolas et al paper used the 'resuspension' method, which might help explain the difference in the kinetics of sporulation

P8 line 2. ...than other similar approaches.

P9 middle. Protease instead of RNase. Please clarify. This is not obvious if one doesn't know the standard protocol. Also

possible just to say 'treated it with protease'.

P10 Is the 'correlation coefficient' just the ratio of amounts in both fractions. Please specify how it was calculated. The sentence suggests that this is specified in materials and methods, but I could not find it there or in the supplementary methods

P10. Fig 5E Are the GO-terms exclusive? Or can one protein be found in several GO terms. Please specify

P12 L8. No GO term was enriched...

P12. What is the difference between cluster 2 and 3 to warrant separating them?

P13, L2,among which is Hfq.

P14, L17 I was surprised to learn that some of the proteins listed (remove italics and capitalize first letters), notably SigE, F, G, K and Mfd are 'characterized RNA binders'. For me they are DNA binding proteins. Please provided references.

P14, L18 Do the authors mean clusters 8 and 9 instead of 7 and 8. If not the paragraph structure is strange and it is difficult to follow how they authors went from discussing 180 proteins to 45.

P14, L23 It took me a long time to realise that the '7 other unknown proteins' were the remaining seven from the 38/45 of the previous paragraph. Please specify

P15. L3. Put 'Except YxnB' at the beginning of the sentence rather than the end.

P15 It is not clear why YhfW is in the table (1) of those proteins with non-enzymatic functions if it is a putative oxido-reductase. Please clarify the distinction

P16 Can the authors rule out the possibility that the P32 signal associated with KhpA and SpoVR in Figs 8B and 9B is not due to phosphorylation of the proteins rather than RNA binding? Have the samples been treated with RNase or phosphatase? If not, please mention this caveat or explain how phosphorylation can be ruled out.

P16, Fig. 8C. In the alphafold model, the polyU sequence forms a helical structure. This seems incorrect. Please clarify.

P17, Fig 9C. Please give confidence score for alphafold model, as was done for KhpA

P19, L11. ...RBPs that bind these RNAs. (there is no evidence for stabilization)

P19, last line. 'possibly', rather than 'probably'

Reviewer #1:

The data in the manuscript are valuable and will provide a useful resource for the community, but the presentation requires some work to make it accessible. The manuscript needs proofreading throughout. Some sections (eg: analysis of clusters) are overly verbose and distract from the key messages. Shortening the manuscript to focus on the key findings will improve readability.

As suggested by the referee, the manuscript has been reworked and shortened to highlight key messages and make it easier to read.

Major comments:

1. Pg 6 second paragraph. The first RBPome for a Gram positive was PMID: 35610211, which is referenced later in the manuscript. I think the author mean the first time OOPS has been applied to a Gram positive.

In the revised version, we have clarified this point (lane 89). Indeed, we meant OOPS was applied for the first time to a Gram+ bacteria.

2. Pg 11-14: 'Classification of RBPome in different clusters'. The authors present a very detailed description of each cluster, which seems unwarranted for clusters 1-7 and becomes an extended list of observations. Clusters 8 and 9 contain sporulation genes and the authors choose KhpA and SpoVR from these clusters. I suggest significantly condensing or reducing clusters 1-6.

The detailed descriptions of each cluster have been significantly condensed and focus now only on highlighting specific points.

3. Pg 14: "Highlight on unknown proteins ..." The authors focus on 7 unknown proteins from clusters 8 and 9. Again, each protein is described in a separate paragraph but there are no conclusions drawn from the list of observations. The authors conclude with "...our results show that these proteins bind RNA during sporulation process..." but it is not clear what results have been presented.

This section has been significantly shortened and revised to emphasize proteins probable involvement in sporulation and/or RNA binding.

4. Pg 16: The data from Abrahamson et al is described in a few sentences and their structure is presented in Figure 8. It might be better to describe the main finding succinctly - KhpA is predicted to require RNA binding for stability - or repeat these analysis and AF3 structure predictions, which would also allow modelling with *B. subtilis* sRNA rather than poly(U).

We performed structure prediction (see below, ipTM = 0,44) using the KhpA homodimer and CsfG, the most abundant sporulation-specific sRNA in *B. subtilis* at 7h in DSM (doi: 10.4161/rna.8.3.14998; doi: 10.1126/science.1206848). Notably, the 4 residues (R61, K52, R55, K74) predicted to form the strongest bonds with CsfG are those identified in the polyU-KhpAx2 complex (added in Fig 8), which increases confidence in these predictions. However, we chose not to include structural models of KhpA in complex with CsfG (or with other sRNAs) in the manuscript, as there is no biological evidence for a KhpA-CsfG interaction *in vivo*, and we wish to avoid suggesting that we have a rationale for proposing such an interaction.

To eliminate the confusing helical conformation of the polyU predicted by AF3 and focus on the KhpA-RNA interaction, we shortened polyU from 43 to 20nt and made a new prediction, thus increasing the ipTM from 0.65 to 0.81. This change is reflected in the revised text (lanes 325-327) and updated figure 8C.

5. Pg 17: khpA is shown to contribute to sporulation. The authors conclude on pg 20 that KhpB (Jag) does not have a specific role in sporulation, but it is not clear what supports this conclusion. Does a khpA jag double deletion have a more pronounced sporulation defect?

We performed sporulation tests on *khpA/khpB* double mutant and *khpB* single mutant. They both have a similar phenotype, showing no significant differences with WT (See figure below). We chose not to add these observations in the manuscript since they are not clarifying the results and will be the object for future investigations. However, we rephrased the sentence about KhpB not being involved in sporulation, to avoid overinterpretation (lane 379).

6. Based on work in *Streptococcus* and *Fusobacterium*, is there any indication of the likely RNA ligands for KhpA in *B. subtilis*? KhpB has been described in *Clostridioides difficile* (PMID: 37223250) that is also sporulating and may provide some clues to the likely ligands.

We had the same rationale than the reviewer about spore formers and had a closer look at *C. difficile* in which KhpB was found to bind several many RNAs including 15 sRNAs, with 3 that are also bound by KhpA: (nc008, nc070 and nc088). These sRNAs do not exhibit sequence homology or genomic context that allowed us to identify potential targets in *B. subtilis* (This is also the case for *Fusobacterium* and *Streptococcus*). It is probable that KhpA recognizes conserved RNA targets, mRNAs or ncRNAs, across species. However, RIP-seq or CRAC experiments in *B. subtilis* will be unavoidable to explore KhpA RNA interactome and its conservation across species and this will be the object of future investigation.

We added a sentence (lanes 372-373): "This raises the question of whether KhpA recognizes conserved RNA targets across species, particularly in spore formers such as *C. difficile*."

7. Pg 17: From the AF3 SpoVR-RNA structure, can the authors infer anything about how the RNA is binding to SpoVR?

We have modified this section (lanes 343-345) and the figure (see Fig9C) for this section: "Interestingly, AlphaFold3 predicted that a poly-U RNA binds SpoVR within a positively charged surface cavity (ipTM = 0.62), despite the absence of a known RNA recognition domain (Fig. 9C).

Minor comments:

1. The manuscript needs careful proofreading for grammatical issues throughout. I have not listed them here as there are too many.

We have thoroughly revised the manuscript for clarity and conciseness, and had it proofread for grammatical issues.

2. The phylum Firmicutes is now Bacillota (PMID: 34694987).

All occurrences in the text and in the table 1 have been changed.

Reviewer #2 (Comments for the Author):

The paper by Kaboré et al describes the large-scale identification of RNA binding proteins in *B. subtilis*, a study that is long-overdue. Using OOPS (orthogonal organic phase separation) the authors identify over a thousand potential RNA binding proteins in both normal growth and sporulation conditions. The paper is well-written, in particular the introduction which sets the paper up well. I have only a few minor comments to improve the presentation of the manuscript.

I felt that a major section of the discussion (from bottom of P17 to top of P19) would have fit better in the results after the presentation of the 10 clusters. A resumé of the highlights of what was actually found will be expected earlier by the reader. It will shorten the discussion, but that's OK

We agree and have moved this part to the Results, condensing and combining it with cluster presentation section.

Please use line-numbers to help review process!

We are really sorry about this and added them in the revised manuscript.

P4 2 lines from bottom. To date, most research has focused

This has been corrected in the revised text.

P6 and Fig 2. Please specify whether all 150 potential sRNAs are included in the heat-map

Thank you for pointing this out. It is now specified in the text and in the legend.

P7. The sentence starting "Those putative sRNAs ..." is long and awkward. Please rephrase
The sentences have been rephrased (lane 106).

P7 middle. I suggest modifying the first sentence of the new section to 'To identify RNA-protein complexes involved in RNA-mediated regulation during sporulation,' to better make the connection with the previous section
The sentence has been rephrased (lane 117).

P7 Please specify that the sporulation method used in this paper is the 'exhaustion' method, for sporulation experts. It could also be mentioned that the Nicolas et al paper used the 'resuspension' method, which might help explain the difference in the kinetics of sporulation

Lanes 124-126: We specified these different protocols in the following sentence: "Compared to the conditions used in (Nicolas et al. 2012), which employed the 'resuspension method', we observed a slight delay in sporulation in our experiments which followed the 'exhaustion protocol' (Fig. 2 and 3A)."

P8 line 2. ...than other similar approaches.
This has been corrected in the text (lane 132).

P9 middle. Protease instead of RNase. Please clarify. This is not obvious if one doesn't know the standard protocol. Also possible just to say 'treated it with protease'.
"instead of RNase" referring to the standard protocol have been removed from the text for clarity (lane 139).

P10 Is the 'correlation coefficient' just the ratio of amounts in both fractions. Please specify how it was calculated. The sentence suggests that this is specified in materials and methods, but I could not find it there or in the supplementary methods

The specifications about correlation coefficient are stated in supplementary material & methods in the proteomics analysis section: "A fraction correlation coefficient was calculated for each protein recovered in OOPS to assess the differences between recovery of one protein on both fractions, according to this formula: Fraction correlation coefficient = $1 - \sqrt{0.25 \times (FP_{LB} - RBP_{LB})^2 + 0.25 \times (FP_{DSM\ 3h} - RBP_{DSM\ 3h})^2 + 0.25 \times (FP_{DSM\ 7h} - RBP_{DSM\ 7h})^2 + 0.25 \times (FP_{DSM\ 12h} - RBP_{DSM\ 12h})^2}$." We added "supplementary Material and Methods" (lane 180).

P10. Fig 5E Are the GO-terms exclusive? Or can one protein be found in several GO terms. Please specify
GO-terms are not exclusive. A sentence has been added in the text (lanes 193-195): "The top 500 proteins identified in the RBP fractions (from all samples combined) were analyzed for Gene-Ontology (GO) terms enrichment. Notably, one protein can be found in several GO-terms (Fig. 5E). "

P12 L8. No GO term was enriched...
This text has been removed due to rephrasing to make it more concise.

P12. What is the difference between cluster 2 and 3 to warrant separating them?
Clustering analysis performed with Cluster3 (DOI: [10.1093/bioinformatics/bth078](https://doi.org/10.1093/bioinformatics/bth078)) led to 10 clusters including cluster 2 and 3 that show highly similar profiles. It seems that the main difference between them is the profile differences between both FP and RBP fractions, highlighted by correlation coefficients that are globally much lower for proteins found in cluster 3 than those from cluster 2. We combined them for description in the text and globally condensed this part of the result section.

P13, L2,among which is Hfq.
This has been rephrased in the revised text (lane 264).

P14, L17 I was surprised to learn that some of the proteins listed (remove italics and capitalize first letters), notably SigE, F, G, K and Mfd are 'characterized RNA binders'. For me they are DNA binding proteins. Please provided references.
We thank the reviewer for this remark. Indeed, our sentence was confusing since we mixed into the brackets both potential and characterized RNA binders, and as there are indeed no report for RNA binding ability of sigma factors and mfd. We chose to remove this sentence.

P14, L18 Do the authors mean clusters 8 and 9 instead of 7 and 8. If not the paragraph structure is strange and it is difficult to follow how they authors went from discussing 180 proteins to 45.
Thank you for pointing this mistake out. Indeed, we meant clusters 8 and 9. This has been corrected in the text (lane 284).

P14, L23 It took me a long time to realise that the '7 other unknown proteins' were the remaining seven from the 38/45 of the previous paragraph. Please specify
The sentence has been rephrased (lanes 288-290).

P15. L3. Put 'Except YxnB' at the beginning of the sentence rather than the end.
The sentence has been rephrased (lane 293).

P15 It is not clear why YhfW is in the table (1) of those proteins with non-enzymatic functions if it is a putative oxido-reductase. Please clarify the distinction

YhfW was not annotated as putative enzyme in Subtiwiki, which initially led us to include it among the unknown proteins. However, further exploration of available data and conservation analysis points towards YhfW being an oxido-reductase. We thus have removed it for the selected unknown proteins. This correction has been made in the main text, Table 1 and Sup Table S7.

P16 Can the authors rule out the possibility that the P32 signal associated with KhpA and SpoVR in Figs 8B and 9B is not due to phosphorylation of the proteins rather than RNA binding? Have the samples been treated with RNase or phosphatase? If not, please mention this caveat or explain how phosphorylation can be ruled out. We did not treat the samples with RNase or phosphatase (but only with DNase) because T4 polynucleotide kinase (PNK) is the only present kinase in the reaction and is specific to nucleic acids (RNA/DNA). Indeed, *Bacillus subtilis* cultures were grown in LB or DSM and subjected to UV to crosslink proteins in direct interaction with RNA; then KhpA-HF or SpoVR-HF were double-purified from *Bacillus subtilis* under successive native (with anti-Flag beads) and denaturing (with Nickel beads) conditions leading to a pure protein. Then radioactive ATP and T4 PNK was added to the pure protein. SpoVR and KhpA are not suspected to be kinases (they neither carry kinase domain nor are detected in all the phosphoproteomes from *B. subtilis*). Then, T4PNK being the sole kinase and not being able to phosphorylate proteins, the P32 signal is associated to radioactive RNA bound to KhpA and SpoVR.

Finally, it is worth noting that OOPS has been performed both with and without UV crosslinking. KhpA was not detected in the non-UV sample while SpoVR is 7X more abundant in the UV-treated sample than in the non-UV samples at DSM7h, consistent with them binding RNA.

P16, Fig. 8C. In the alphafold model, the polyU sequence forms a helical structure. This seems incorrect. Please clarify.

To avoid this confusing helical structure predicted by AF3, we shortened the polyU from 43 to 20nt and made a new prediction, thus increasing the ipTM from 0.65 to 0.81. This has been changed in the text (lanes 325-327). The figure 8C has been modified.

P17, Fig 9C. Please give confidence score for alphafold model, as was done for KhpA

This has been added in the text as long as more details for the RNA-protein interaction (lanes 325-327).

P19, L11. ...RBPs that bind these RNAs. (there is no evidence for stabilization)

This has been rephrased in the revised text.

P19, last line. 'possibly', rather than 'probably'

This has been rephrased.

Re: mSystems00496-25R1 (Remodeling of RNA-Binding Proteome and RNA-mediated regulation as a new layer of control of sporulation)

Dear Dr. Clementine Delan-Forino:

I am pleased to report that both reviewers were happy with all of your additions, modifications, and responses. The flow of the manuscript is very much improved. However, I would ask that you go through the manuscript one more time to look for missing articles, agreement, ensure tense is consistent, etc. Feel free to contact me if you have any questions or if I can help. I have attached a version of the main text with a few things highlighted.

Revision Guidelines

Sincerely,
Sarah Svensson
Editor
mSystems

Reviewer #1 (Comments for the Author):

The authors have addressed my comments.

Reviewer #2 (Comments for the Author):

I am happy with the changes made to the manuscript

Reviewer #1:

The data in the manuscript are valuable and will provide a useful resource for the community, but the presentation requires some work to make it accessible. The manuscript needs proofreading throughout. Some sections (eg: analysis of clusters) are overly verbose and distract from the key messages. Shortening the manuscript to focus on the key findings will improve readability.

As suggested by the referee, the manuscript has been reworked and shortened to highlight key messages and make it easier to read.

Major comments:

1. Pg 6 second paragraph. The first RBPome for a Gram positive was PMID: 35610211, which is referenced later in the manuscript. I think the author mean the first time OOPS has been applied to a Gram positive.

In the revised version, we have clarified this point (lane 89). Indeed, we meant OOPS was applied for the first time to a Gram+ bacteria.

2. Pg 11-14: 'Classification of RBPome in different clusters'. The authors present a very detailed description of each cluster, which seems unwarranted for clusters 1-7 and becomes an extended list of observations. Clusters 8 and 9 contain sporulation genes and the authors choose KhpA and SpoVR from these clusters. I suggest significantly condensing or reducing clusters 1-6.

The detailed descriptions of each cluster have been significantly condensed and focus now only on highlighting specific points.

3. Pg 14: "Highlight on unknown proteins ..." The authors focus on 7 unknown proteins from clusters 8 and 9. Again, each protein is described in a separate paragraph but there are no conclusions drawn from the list of observations. The authors conclude with "...our results show that these proteins bind RNA during sporulation process..." but it is not clear what results have been presented.

This section has been significantly shortened and revised to emphasize proteins probable involvement in sporulation and/or RNA binding.

4. Pg 16: The data from Abrahamson et al is described in a few sentences and their structure is presented in Figure 8. It might be better to describe the main finding succinctly - KhpA is predicted to require RNA binding for stability - or repeat these analysis and AF3 structure predictions, which would also allow modelling with *B. subtilis* sRNA rather than poly(U).

We performed structure prediction (see below, ipTM = 0,44) using the KhpA homodimer and CsfG, the most abundant sporulation-specific sRNA in *B. subtilis* at 7h in DSM (doi: 10.4161/rna.8.3.14998; doi: 10.1126/science.1206848). Notably, the 4 residues (R61, K52, R55, K74) predicted to form the strongest bonds with CsfG are those identified in the polyU-KhpAx2 complex (added in Fig 8), which increases confidence in these predictions. However, we chose not to include structural models of KhpA in complex with CsfG (or with other sRNAs) in the manuscript, as there is no biological evidence for a KhpA-CsfG interaction *in vivo*, and we wish to avoid suggesting that we have a rationale for proposing such an interaction.

To eliminate the confusing helical conformation of the polyU predicted by AF3 and focus on the KhpA-RNA interaction, we shortened polyU from 43 to 20nt and made a new prediction, thus increasing the ipTM from 0.65 to 0.81. This change is reflected in the revised text (lanes 325-327) and updated figure 8C.

5. Pg 17: khpA is shown to contribute to sporulation. The authors conclude on pg 20 that KhpB (Jag) does not have a specific role in sporulation, but it is not clear what supports this conclusion. Does a khpA jag double deletion have a more pronounced sporulation defect?

We performed sporulation tests on *khpA/khpB* double mutant and *khpB* single mutant. They both have a similar phenotype, showing no significant differences with WT (See figure below). We chose not to add these observations in the manuscript since they are not clarifying the results and will be the object for future investigations. However, we rephrased the sentence about KhpB not being involved in sporulation, to avoid overinterpretation (lane 379).

6. Based on work in *Streptococcus* and *Fusobacterium*, is there any indication of the likely RNA ligands for KhpA in *B. subtilis*? KhpB has been described in *Clostridioides difficile* (PMID: 37223250) that is also sporulating and may provide some clues to the likely ligands.

We had the same rationale than the reviewer about spore formers and had a closer look at *C. difficile* in which KhpB was found to bind several many RNAs including 15 sRNAs, with 3 that are also bound by KhpA: (nc008, nc070 and nc088). These sRNAs do not exhibit sequence homology or genomic context that allowed us to identify potential targets in *B. subtilis* (This is also the case for *Fusobacterium* and *Streptococcus*). It is probable that KhpA recognizes conserved RNA targets, mRNAs or ncRNAs, across species. However, RIP-seq or CRAC experiments in *B. subtilis* will be unavoidable to explore KhpA RNA interactome and its conservation across species and this will be the object of future investigation.

We added a sentence (lanes 372-373): "This raises the question of whether KhpA recognizes conserved RNA targets across species, particularly in spore formers such as *C. difficile*."

7. Pg 17: From the AF3 SpoVR-RNA structure, can the authors infer anything about how the RNA is binding to SpoVR?

We have modified this section (lanes 343-345) and the figure (see Fig9C) for this section: "Interestingly, AlphaFold3 predicted that a poly-U RNA binds SpoVR within a positively charged surface cavity (ipTM = 0.62), despite the absence of a known RNA recognition domain (Fig. 9C).

Minor comments:

1. The manuscript needs careful proofreading for grammatical issues throughout. I have not listed them here as there are too many.

We have thoroughly revised the manuscript for clarity and conciseness, and had it proofread for grammatical issues.

2. The phylum Firmicutes is now Bacillota (PMID: 34694987).

All occurrences in the text and in the table 1 have been changed.

Reviewer #2 (Comments for the Author):

The paper by Kaboré et al describes the large-scale identification of RNA binding proteins in *B. subtilis*, a study that is long-overdue. Using OOPS (orthogonal organic phase separation) the authors identify over a thousand potential RNA binding proteins in both normal growth and sporulation conditions. The paper is well-written, in particular the introduction which sets the paper up well. I have only a few minor comments to improve the presentation of the manuscript.

I felt that a major section of the discussion (from bottom of P17 to top of P19) would have fit better in the results after the presentation of the 10 clusters. A resumé of the highlights of what was actually found will be expected earlier by the reader. It will shorten the discussion, but that's OK

We agree and have moved this part to the Results, condensing and combining it with cluster presentation section.

Please use line-numbers to help review process!

We are really sorry about this and added them in the revised manuscript.

P4 2 lines from bottom. To date, most research has focused

This has been corrected in the revised text.

P6 and Fig 2. Please specify whether all 150 potential sRNAs are included in the heat-map

Thank you for pointing this out. It is now specified in the text and in the legend.

P7. The sentence starting "Those putative sRNAs ..." is long and awkward. Please rephrase
The sentences have been rephrased (lane 106).

P7 middle. I suggest modifying the first sentence of the new section to 'To identify RNA-protein complexes involved in RNA-mediated regulation during sporulation,' to better make the connection with the previous section
The sentence has been rephrased (lane 117).

P7 Please specify that the sporulation method used in this paper is the 'exhaustion' method, for sporulation experts. It could also be mentioned that the Nicolas et al paper used the 'resuspension' method, which might help explain the difference in the kinetics of sporulation

Lanes 124-126: We specified these different protocols in the following sentence: "Compared to the conditions used in (Nicolas et al. 2012), which employed the 'resuspension method', we observed a slight delay in sporulation in our experiments which followed the 'exhaustion protocol' (Fig. 2 and 3A)."

P8 line 2. ...than other similar approaches.
This has been corrected in the text (lane 132).

P9 middle. Protease instead of RNase. Please clarify. This is not obvious if one doesn't know the standard protocol. Also possible just to say 'treated it with protease'.
"instead of RNase" referring to the standard protocol have been removed from the text for clarity (lane 139).

P10 Is the 'correlation coefficient' just the ratio of amounts in both fractions. Please specify how it was calculated. The sentence suggests that this is specified in materials and methods, but I could not find it there or in the supplementary methods

The specifications about correlation coefficient are stated in supplementary material & methods in the proteomics analysis section: "A fraction correlation coefficient was calculated for each protein recovered in OOPS to assess the differences between recovery of one protein on both fractions, according to this formula: Fraction correlation coefficient = $1 - \sqrt{0.25 \times (FP_{LB} - RBP_{LB})^2 + 0.25 \times (FP_{DSM\ 3h} - RBP_{DSM\ 3h})^2 + 0.25 \times (FP_{DSM\ 7h} - RBP_{DSM\ 7h})^2 + 0.25 \times (FP_{DSM\ 12h} - RBP_{DSM\ 12h})^2}$." We added "supplementary Material and Methods" (lane 180).

P10. Fig 5E Are the GO-terms exclusive? Or can one protein be found in several GO terms. Please specify
GO-terms are not exclusive. A sentence has been added in the text (lanes 193-195): "The top 500 proteins identified in the RBP fractions (from all samples combined) were analyzed for Gene-Ontology (GO) terms enrichment. Notably, one protein can be found in several GO-terms (Fig. 5E). "

P12 L8. No GO term was enriched...
This text has been removed due to rephrasing to make it more concise.

P12. What is the difference between cluster 2 and 3 to warrant separating them?
Clustering analysis performed with Cluster3 (DOI: [10.1093/bioinformatics/bth078](https://doi.org/10.1093/bioinformatics/bth078)) led to 10 clusters including cluster 2 and 3 that show highly similar profiles. It seems that the main difference between them is the profile differences between both FP and RBP fractions, highlighted by correlation coefficients that are globally much lower for proteins found in cluster 3 than those from cluster 2. We combined them for description in the text and globally condensed this part of the result section.

P13, L2,among which is Hfq.
This has been rephrased in the revised text (lane 264).

P14, L17 I was surprised to learn that some of the proteins listed (remove italics and capitalize first letters), notably SigE, F, G, K and Mfd are 'characterized RNA binders'. For me they are DNA binding proteins. Please provided references.
We thank the reviewer for this remark. Indeed, our sentence was confusing since we mixed into the brackets both potential and characterized RNA binders, and as there are indeed no report for RNA binding ability of sigma factors and mfd. We chose to remove this sentence.

P14, L18 Do the authors mean clusters 8 and 9 instead of 7 and 8. If not the paragraph structure is strange and it is difficult to follow how they authors went from discussing 180 proteins to 45.
Thank you for pointing this mistake out. Indeed, we meant clusters 8 and 9. This has been corrected in the text (lane 284).

P14, L23 It took me a long time to realise that the '7 other unknown proteins' were the remaining seven from the 38/45 of the previous paragraph. Please specify
The sentence has been rephrased (lanes 288-290).

P15. L3. Put 'Except YxnB' at the beginning of the sentence rather than the end.
The sentence has been rephrased (lane 293).

P15 It is not clear why YhfW is in the table (1) of those proteins with non-enzymatic functions if it is a putative oxido-reductase. Please clarify the distinction

YhfW was not annotated as putative enzyme in Subtiwiki, which initially led us to include it among the unknown proteins. However, further exploration of available data and conservation analysis points towards YhfW being an oxido-reductase. We thus have removed it for the selected unknown proteins. This correction has been made in the main text, Table 1 and Sup Table S7.

P16 Can the authors rule out the possibility that the P32 signal associated with KhpA and SpoVR in Figs 8B and 9B is not due to phosphorylation of the proteins rather than RNA binding? Have the samples been treated with RNase or phosphatase? If not, please mention this caveat or explain how phosphorylation can be ruled out. We did not treat the samples with RNase or phosphatase (but only with DNase) because T4 polynucleotide kinase (PNK) is the only present kinase in the reaction and is specific to nucleic acids (RNA/DNA). Indeed, *Bacillus subtilis* cultures were grown in LB or DSM and subjected to UV to crosslink proteins in direct interaction with RNA; then KhpA-HF or SpoVR-HF were double-purified from *Bacillus subtilis* under successive native (with anti-Flag beads) and denaturing (with Nickel beads) conditions leading to a pure protein. Then radioactive ATP and T4 PNK was added to the pure protein. SpoVR and KhpA are not suspected to be kinases (they neither carry kinase domain nor are detected in all the phosphoproteomes from *B. subtilis*). Then, T4PNK being the sole kinase and not being able to phosphorylate proteins, the P32 signal is associated to radioactive RNA bound to KhpA and SpoVR.

Finally, it is worth noting that OOPS has been performed both with and without UV crosslinking. KhpA was not detected in the non-UV sample while SpoVR is 7X more abundant in the UV-treated sample than in the non-UV samples at DSM7h, consistent with them binding RNA.

P16, Fig. 8C. In the alphafold model, the polyU sequence forms a helical structure. This seems incorrect. Please clarify.

To avoid this confusing helical structure predicted by AF3, we shortened the polyU from 43 to 20nt and made a new prediction, thus increasing the ipTM from 0.65 to 0.81. This has been changed in the text (lanes 325-327). The figure 8C has been modified.

P17, Fig 9C. Please give confidence score for alphafold model, as was done for KhpA

This has been added in the text as long as more details for the RNA-protein interaction (lanes 325-327).

P19, L11. ...RBPs that bind these RNAs. (there is no evidence for stabilization)

This has been rephrased in the revised text.

P19, last line. 'possibly', rather than 'probably'

This has been rephrased.

Re: mSystems00496-25R2 (Remodeling of RNA-Binding Proteome and RNA-mediated regulation as a new layer of control of sporulation)

Dear Dr. Clementine Delan-Forino:

Thank you very for addressing my final minor concerns.

I am happy to report that your manuscript has been accepted, and I am forwarding it to the ASM production staff for publication. Your paper will first be checked to make sure all elements meet the technical requirements. ASM staff will contact you if anything needs to be revised before copyediting and production can begin. Otherwise, you will be notified when your proofs are ready to be viewed.

Cover Image Submissions: If you would like to submit a potential Cover Image, please email a file and a short legend to mSystems@asmusa.org. Please note that we can only consider images that (i) the authors created or own and (ii) have not been previously published. By submitting, you agree that the image can be used under the same terms as the published article. Image File requirements: TIF/EPS, 7.5 inches wide by 8.25 inches tall (at least 2,250 pixels wide by 2,475 pixels tall), minimum 300 dpi resolution (600 dpi preferred), RGB, and no figure elements, e.g., arrows or panel labels. The legend should be a short description of the image, 1-2 sentences recommended. Please download and use this interactive template in Adobe to ensure that your proposed cover image meets our size requirements (<https://journals.asm.org/pb-assets/pdf-text-excel-files/ASM-Interactive-Sizing-Cover-Template-1715689791.pdf>).

Sincerely,
Sarah Svensson
Editor
mSystems